# Association of Fatty Liver Index with Incident Diabetes Risk in Patients Initiating Statin–Therapy: A 6-Year Retrospective Study

**DOI:** 10.3390/diagnostics13030503

**Published:** 2023-01-30

**Authors:** Georgia Anastasiou, Evangelos Liberopoulos, Ermioni Petkou, Amalia Despoina Koutsogianni, Petros Spyridwnas Adamidis, George Liamis, Evangelia Ntzani, Fotios Barkas

**Affiliations:** 1Department of Internal Medicine, Faculty of Medicine, School of Health Sciences, University of Ioannina, 451 10 Ioannina, Greece; 21st Propaedeutic Department of Medicine, Laiko General Hospital of Athens, School of Medicine, National and Kapodistrian University of Athens, 157 72 Athens, Greece; 3Department of Hygiene & Epidemiology, Faculty of Medicine, School of Health Sciences, University of Ioannina, 451 10 Ioannina, Greece

**Keywords:** non-alcoholic fatty liver disease, fatty liver index, diabetes, statin, metabolic syndrome

## Abstract

Background: Statins are associated with new-onset type 2 diabetes (T2D), mainly in patients with metabolic syndrome (MetS). The fatty liver index (FLI) is used as a prognostic score for the diagnosis of non-alcoholic fatty liver disease (NAFLD), which is common in patients with MetS. We aimed to investigate the association of FLI with new-onset T2D in patients initiating statin therapy. Methods: A retrospective observational study including 1241 individuals with dyslipidemia and followed up for ≥3 years. Patients with T2D and those receiving lipid-lowering treatment at the baseline visit were excluded. Models with clinical and laboratory parameters were used to assess the association of FLI with incident T2D. Results: Among the 882 eligible subjects, 11% developed T2D during the follow-up (6 years; IQR: 4–10 years). After adjusting for sex, age and MetS parameters, a multivariate analysis revealed that age (HR:1.05; 95%CI: 1.01–1.09, *p* < 0.05), fasting plasma glucose (HR: 1.09; 95%CI: 1.06–1.13, *p* < 0.001) and FLI (HR: 1.02; 95%CI: 1.01–1.04, *p* < 0.01) were independently associated with T2D risk. The subjects with probable NAFLD (FLI ≥ 60) had a three-fold increased T2D risk compared with the subjects with FLI < 60 (HR: 3.14; 95%CI: 1.50–6.59, *p* = 0.001). A ROC curve analysis showed that FLI had a significant, although poor, predictive value for assessing T2D risk (C-Statistic: 0.67; 95%CI: 0.58–0.77, *p* = 0.001). Higher FLI values were associated with reduced T2D-free survival (log-rank = 15.46, *p* < 0.001). Conclusions: FLI is significantly and independently associated with new-onset T2D risk in patients initiating statin therapy.

## 1. Introduction

Non-alcoholic fatty liver disease (NAFLD) is the term used for a range of conditions caused by fat accumulation in the liver in the absence of increased alcohol consumption [1,2,3]. NAFLD is the most common chronic liver disease, with a worldwide prevalence of 32.4% (95% confidence intervals, CI: 29.9–34.9%) [4]. Apart from being the major cause of cirrhosis in the developed countries, the accumulating evidence suggests that NAFLD is the hepatic component of metabolic syndrome (MetS) and is associated with new-onset type 2 diabetes (T2D) and atherosclerotic cardiovascular disease (ASCVD) [4,5,6,7,8]. NAFLD and T2D share common pathophysiological mechanisms, with insulin resistance being the primary one [9]. It is well established that metabolic hepatic steatosis is a phenotype that includes diverse and complex causes, as well as demonstrating a wide spectrum of clinical severity, but also a considerable inter-patient variability [9]. In this context, “MAFLD” (metabolic-dysfunction-associated fatty liver disease) is preferred over ‘’NAFLD’’ as a more appropriate general term [9]. Although liver biopsy remains the gold standard, various prognostic models have been developed for NAFLD diagnosis, such as the fatty liver index (FLI) [10]. On the other hand, statins are the cornerstone therapy in ASCVD prevention and might have a role in the treatment of NAFLD [11,12,13]. However, statins have been associated with increased T2D risk, mostly in individuals with MetS components, such as prediabetes, obesity, hypertension, and increased triglycerides [14,15,16,17,18]. Thus, there is a need for identifying biomarkers predicting statin-induced T2D.

In this context, the aim of the present study was to investigate the association of FLI with new-onset T2D in patients initiating statin therapy and followed up for ≥3 years.

## 2. Materials and Methods

As previously described, this was a retrospective study including adult patients with dyslipidemia who attended the Outpatient Lipid Clinic of the University Hospital of Ioannina in Greece for ≥3 years [18,19,20,21]. The study protocol was approved by the local institutional ethics committee (No. 3231, 5 July 2012) and informed consent was obtained from each patient.

All the study participants were Caucasians. Lipid-lowering drug-naïve patients with dyslipidemia were considered eligible for this study. The patients with T2D at baseline and those not prescribed with statin therapy during the follow-up were excluded. A complete assessment of the clinical and laboratory profile was performed at the baseline visit by a physician, including: (i) sex and age, (ii) body mass index (BMI) and waist circumference, (iii) blood pressure, (iv) fasting plasma glucose (FPG) levels, (v) lipid profile, including total cholesterol (TCHOL), triglycerides (TG), high-(HDL-C) and low-density lipoprotein cholesterol (LDL-C), as well as (vi) liver enzymes, including aspartate aminotransferase (AST), alanine aminotransferase (ALT), gamma-glutamyl transferase (gGT), alkaline phosphatase (ALP) and total and direct bilirubin. The insulin levels were available in only few patients. The office blood pressure measurements were performed with a validated upper-arm cuff BP measurement device and an appropriate cuff size. The corresponding instruments measuring the subjects’ waist, height and weight were validated and calibrated. The waist circumference was measured by placing a tape measure mid-way between the lower border of the costal margin and the top border of the iliac crest. BMI was calculated as: (weight, kg)/(height, m)^2^. During the study conduct, the glycated hemoglobin (HbA1c) was not routinely measured before the diagnosis of T2D, according to national guidelines [22].

Any concomitant therapy was recorded with a particular emphasis on lipid-lowering drugs. The classification of statin intensity as high, moderate or low was based on the average expected LDL-C reduction by ≥50%, 30–50% and <30%, respectively [13]. High-intensity statin therapy includes atorvastatin 40–80 mg and rosuvastatin 20–40 mg, whereas moderate-intensity statin therapy comprises atorvastatin 10–20 mg, rosuvastatin 5–10 mg, simvastatin 20–40 mg, pravastatin 40–80 mg, fluvastatin 40–80 mg and pitavastatin 1–4 mg [13].

T2D was diagnosed by a physician in the case of FPG levels ≥126 mg/dL (6.9 mmol/L) in two separate measurements in different visits, or when the FPG levels were ≥200 mg/dL (11.1 mmol/L) 2 h following 75 g of oral glucose [23]. Overweight and obesity were defined as BMI 25–30 kg/m^2^ and ≥30 kg/m^2^, respectively. The estimation of FLI was based on the following equation: FLI = e^y^/(1 + e^y^) × 100, where y = 0.953 × ln(TG, mg/dL) + 0.139 × BMI, kg/m^2^ + 0.718 × ln(gGT, U/L) + 0.053 × waist circumference, cm–15.745 [24,25]. The proposed cut-off for probable NAFLD is FLI ≥ 60 [24,25].

### Statistical Analysis

The continuous variables were tested for normality by the Kolmogorov–Smirnov test. The data are presented as the mean ± standard deviation (SD) and median [interquartile range (IQR)] for the parametric and non-parametric data, respectively. For the categorical values, frequency counts and percentages were applied. A chi-square test was performed for the interactions between the categorical values. An independent sample *t*-test (parametric and non-parametric) was used for the comparison of the continuous numeric values between two groups. A univariate Cox regression analysis was performed to investigate the association of a factor with the investigated outcome of interest during the follow-up. A multivariate logistic regression analysis was conducted using the variables that were statistically significant in the univariate analyses (the backward conditional method was used). The associations with the outcomes of interest are expressed as hazard ratios (HR), with an accompanying 95% confidence interval (CI). A receiver-operating characteristics (ROC) curve analysis was used to analyze the prognostic value of the investigated variables for the corresponding outcomes of interest. The C-statistic (area under the curve) is presented as a unified estimate of sensitivity and specificity. A Kaplan–Meier analysis was performed to study the differences in event-free survival from the investigated outcomes of interest. Two-tailed significance was defined as *p* < 0.05. The analyses were performed with the Statistical Package for Social Sciences (SPSS) v25.0 software (SPSS Statistics for Windows, Version 28.0. Armonk, New York, NY, USA: IBM Corp).

## 3. Results

Of the initial study participants (*n* = 1241), 882 subjects were eligible for the present study (Figure 1): 57% were males; their median age was 55 (47–63) years and median BMI was (27.2 (24.8–29.9) kg/m^2^); 57% had hypertension; 38% fulfilled the criteria of metabolic syndrome; and 27% were assigned to high-intensity statin therapy. During the follow-up (6 years; IQR: 4–10 years), 11% of the study participants developed T2D.

The study participants’ baseline characteristics according to the FLI cut-off value are shown in Table 1. A lower proportion of patients with FLI ≥ 60 were females, when compared with those with FLI < 60 (52 vs. 66%, *p* < 0.05), and presented with a higher prevalence of MetS components compared with those with FLI < 60, namely hypertension (67 vs. 54%, *p* < 0.05), increased BMI (30.7 (28.9–33.2) vs. 25.9 (23.9–28.2) kg/m^2^, *p* < 0.05), FPG (95 (89–102) vs. 91 (86–99) mg/dL, *p* < 0.05), TG (170 (125–227) vs. (108 (83–144) mg/dL, *p* < 0.05) and low HDL-C (49 (44–58) vs. 57 (47–69) mg/dL, *p* < 0.05) (Table 1).

The univariate Cox regression analysis demonstrated that age, SBP, waist circumference, FPG and FLI, a personal history of hypertension and metabolic syndrome, as well as a family history of T2D, were associated with incident T2D (Table 2). After adjusting for these factors, the multivariate analysis showed that FLI was independently and significantly associated with incident T2D (HR: 1.02, 95% CI: 1.01–1.04, *p* < 0.01), along with age (HR: 1.05, 95% CI: 1.01–1.09, *p* < 0.05) and FPG (HR: 1.09, 95% CI: 1.06–1.13, *p* < 0.001).

The subjects with FLI ≥ 60 had a three-fold increased risk of new-onset T2D compared with those with FLI < 60 (HR: 3.14; 95% CI: 1.50–6.59, *p* < 0.01). The ROC curve analysis showed that FLI had a significant, but poor, predictive value (C-statistic < 0.7) for new-onset T2D (C-Statistic: 0.67; 95% CI: 0.58–0.77, *p* = 0.001; Figure 2). The sensitivity and specificity values at the cut-off of 60 were 61.8% and 63.0%, respectively.

The patients with FLI ≥60 exhibited reduced T2M-free survival (log-rank = 15.49, *p* < 0.001; Figure 3).

## 4. Discussion

To the best of our knowledge, this is the first study to investigate the association between FLI and incident T2D risk in patients initiating statin therapy. The FLI was significantly and independently associated with the risk of new-onset T2D and showed a significant predictive value for the development of T2D in patients initiating statin therapy.

NAFLD is an umbrella term that encompasses a disease continuum ranging from steatosis to steatohepatitis fibrosis and cirrhosis [26]. NAFLD has been associated with ASCVD, MetS, T2D, obesity, atherogenic dyslipidemia and hypertension [4,27,28,29]. Genetic factors, including a variant of the PNPLA3 gene, epigenetic mechanisms and gut microbiome, may also play a role in NAFLD pathogenesis [30,31]. A liver biopsy remains the gold standard for NAFLD diagnosis, but it remains an invasive method with complications [1,32]. Thus, liver ultrasonography and biomarkers, such as the FLI, SteatoTest and NAFLD fat score, are commonly used [1].

Our results are in agreement with past evidence indicating an association between the FLI and risk of incident T2D in the general population [33,34,35,36,37]. A population-based cohort study conducted in Korea and including 5,254,786 young adults aged 20–39 years demonstrated that the risk of new T2D was higher in those with FLI ≥ 60 than the control group (adjusted HR: 4.97; 95% CI: 4.90–5.05) [33]. Likewise, a prospective study of 16,648 participants with prediabetes demonstrated that high FLI was independently associated with the risk of T2D onset after 5 years of follow-up (adjusted HR: 6.87; 95% CI: 5.87–8.05 for men, and HR: 5.80 95% CI: 4.86–6.93 for women) [35]. A PREDAPS study, a prospective cohort including 1142 adults with prediabetes, also showed that hepatic steatosis, as defined by FLI ≥ 60, was independently associated with T2D incidence after 3 years of follow-up (HR: 3.21; 95% CI: 1.45–7.09 in the fully adjusted model) [36]. Similar results were found by a prospective study including 1792 men without MetS showing a higher T2D risk in subjects with FLI ≥ 60 during a mean follow-up of 19 years (HR: 3.19; 95% CI: 2.26–4.52) [37].

Although new-onset T2D is a well-established side effect of statins, their cardiovascular benefit overweighs this risk, especially in individuals at very high and high cardiovascular risk [38,39,40]. The risk factors for statin-associated T2D include high-intensity statin therapy and MetS components, such as overweight/obesity, prediabetes, hypertension and atherogenic dyslipidemia [13,15,16,17,18]. An analysis of TNT (Treating to New Targets), IDEAL (Incremental Decrease in End Points Through Aggressive Lipid Lowering), and SPARCL (Stroke Prevention by Aggressive Reduction in Cholesterol Levels) trials was the first to demonstrate that elevations of baseline FPG, BMI, fasting TG and hypertension increased the risk of new-onset T2D [41]. Similar findings were also noted in a previous analysis of ours, demonstrating that impaired FPG (OR: 6.56, 95% CI: 3.53–12.18, *p* < 0.01), overweight/obesity (OR: 2.65, 95% CI: 1.39–5.05, *p* < 0.01) and mixed dyslipidemia (OR: 3.27, 95 % CI: 1.50–7.15, *p* < 0.01) were associated with incident T2D [16].

The accumulating evidence indicates that NAFLD is associated with T2D onset independently of its components [5]. A meta-analysis included 33 studies with 501,022 individuals (30.8% with NAFLD, diagnosed by imaging techniques or biopsy) of whom 27,953 developed T2D over a median 5 years (IQR: 4.0–19 years) [42]. The patients with NAFLD had a higher risk of incident T2D than those without NAFLD (HR: 2.19, 95% CI: 1.93–2.48) [42]. The patients with more ‘severe’ NAFLD were also more likely to develop incident T2D (HR 2.69, 95% CI: 2.08–3.49), whereas this risk markedly increased across the severity of liver fibrosis (HR: 3.42, 95% CI: 2.29–5.11) [42]. Of note, all the risks were independent of confounding factors, namely age, sex, adiposity measures and other common metabolic risk factors [42]. This evidence is in accordance with the results of the present study, showing an independent association of FLI with T2D onset in statin-treated patients [42].

This study provides a simplified tool for physicians to identify individuals who are at high risk for new T2D following statin treatment. Therefore, routine FLI calculation in patients with dyslipidemia could be useful in clinical practice, because a high FLI points to further evaluation for NAFLD, but also to preventive strategies to ameliorate the statin-associated T2D risk. In this context, a more frequent monitor of glycemic status, along with a more intensive lifestyle intervention aiming at weight reduction, could probably delay both T2D onset and NAFLD progression [43,44]. At the same time, statin therapy should not be withdrawn but initiated according to the guidelines. Physicians should keep in mind that the sooner they initiate statin therapy, the higher cardiovascular benefit they will accomplish, but also note that NAFLD is not a contradiction to statin therapy [2,13,45]. On the contrary, even patients with steatohepatitis and compensated cirrhosis can be safely treated with statins when recommended [2].

The major limitations of this study are its sample size and retrospective design, along with the lack of data on dietary habits, physical activity and ultrasonography- or biopsy-diagnosed NAFLD. Also, the patients were not systematically screened for viral hepatitis at baseline. Moreover, HbA1c was not routinely measured in the non-diabetic individuals during the conduct of this study. Thus, it is possible that a few patients with T2D at the baseline visit could have been included in the present analysis. However, this study had a long follow-up and is the first to evaluate the clinical utility of FLI in the association and prediction of new T2D in statin-treated patients. Although FLI was strongly associated with incident T2D, its predictive value was poor. In this context, larger prospective studies are needed to further investigate this association in statin-treated individuals.

## 5. Conclusions

FLI is significantly and independently associated with incident T2D risk in statin-treated patients. FLI could be a useful tool, not only for identifying probable NAFLD, but also for predicting new T2D in patients initiating statin therapy.

## Figures and Tables

**Figure 1 diagnostics-13-00503-f001:**
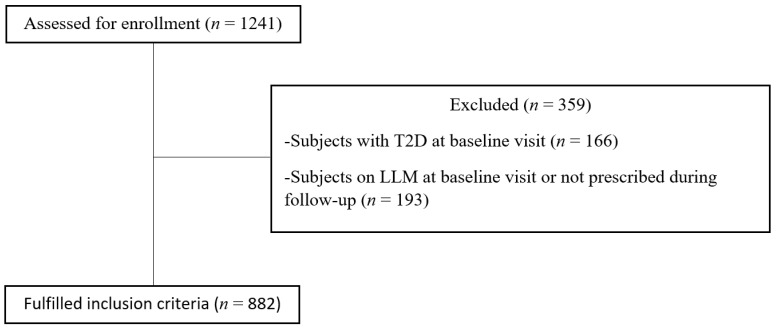
Study flowchart. T2D, type 2 diabetes; LLM, lipid-lowering medication.

**Figure 2 diagnostics-13-00503-f002:**
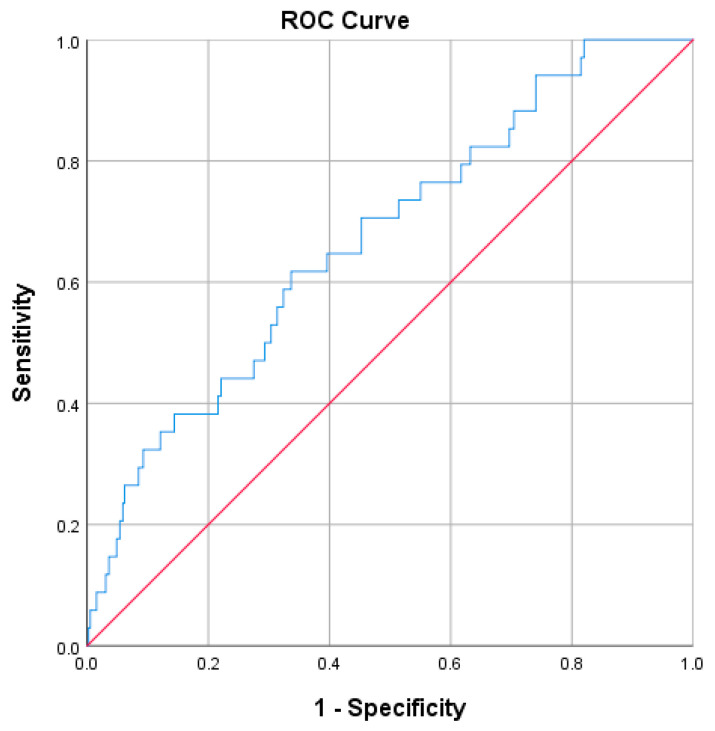
Receiver-operating characteristic (ROC) curve of fatty liver index (FLI) for the prediction of type 2 diabetes. C-Statistic (95% confidence interval): 0.67 (0.58–0.77), *p* < 0.01.

**Figure 3 diagnostics-13-00503-f003:**
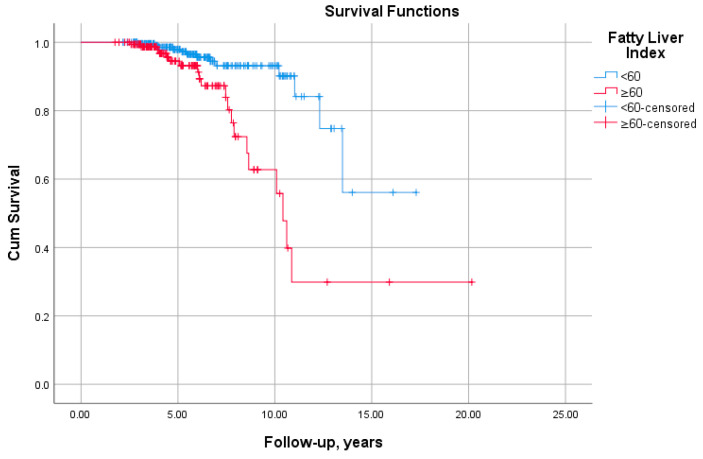
Kaplan–Meier survival curve as stratified by fatty liver index (FLI). FLI ≥ 60 was associated with significantly reduced event-free survival from type 2 diabetes (log-rank = 15.46, *p* = 0.001).

**Table 1 diagnostics-13-00503-t001:** Baseline study participants’ characteristics according to FLI cut-off value.

	Subjects with FLI < 60	Subjects with FLI ≥ 60
N	538	344
Gender (female), %	66	52 *
Age, years	55 (49–63)	59 (50–64)
Smoking, %	18	17
Atherosclerotic cardiovascular disease, %	7	10
Metabolic syndrome, %	25	68 *
Hypertension, %	54	67 *
Family history of T2D, %	20	18
SBP, mmHg	139 (122–150)	140 (130–150)
DBP, mmHg	85 (80–95)	90 (80–94)
BMI, kg/m^2^	25.9 (23.9–28.2)	30.7 (28.9–33.2) *
Waist circumference, cm	91 ± 9	105 ± 10
FPG, mg/dL	91 (86–99)	95 (89–102) *
TCHOL, mg/dL	259 (230–290)	253 (228–284)
TG, mg/dL	108 (83–144)	170 (125-227) *
HDL-C, mg/dL	57 (47–69)	49 (44–58) *
LDL-C, mg/dL	176 (151–200)	168 (144–190) *
Assigned lipid-lowering treatment		
High-intensity statin, %	27	19
Ezetimibe, %	18	15
Fibrates, %	0	8 *
Omega-3 fatty acids, %	2	6 *
Coleveselam, %	1	1

* *p* < 0.05, for the comparison between groups. Values are expressed as median (IQR), unless percentages are shown. Abbreviations: BMI, body mass index; DBP, diastolic blood pressure; FLI, fatty liver index; FPG, fasting plasma glucose; HDL-C, high-density lipoprotein cholesterol; LDL-C, low-density lipoprotein cholesterol; IQR, interquartile range; SBP, systolic blood pressure; T2D, type 2 diabetes; TCHOL, total cholesterol; TG, triglycerides.

**Table 2 diagnostics-13-00503-t002:** Univariate and multivariate analyses of factors associated with incident type 2 diabetes.

Variables	Univariate Analysis	Multivariate Analysis
Age, per 1-year increase	1.06 (1.04–1.08), *p* < 0.001	1.05 (1.01–1.09), *p* < 0.05
Hypertension	1.67 (1.06–2.64), *p* < 0.05	-
Metabolic syndrome	4.35 (2.67–7.09), *p* < 0.001	-
Family history of T2D	2.95 (1.79–4.87), *p* < 0.001	-
SBP, per 1 mm Hg increase	1.02 (1.01–1.03), *p* = 0.001	-
Waist, per 1 cm increase	1.05 (1.02–1.08), *p* < 0.001	-
FPG, per 1 mg/dL increase	1.03 (1.02–1.04), *p* < 0.001	1.09 (1.06–1.13), *p* < 0.001
Fatty liver index, per 1-unit increased	1.03 (1.02–1.05), *p* < 0.001	1.02 (1.01–1.04), *p* < 0.01

The results are expressed as HR (95% CI). Abbreviations: FPG, fasting plasma glucose; HR, hazard ratio; SBP, systolic blood pressure; T2D, type 2 diabetes; 95% CI, 95% confidence interval.

## Data Availability

The data are unavailable due to privacy restrictions.

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
