# Peer review of "Association of Fatty Liver Index with Incident Diabetes Risk in Patients Initiating Statin–Therapy: A 6-Year Retrospective Study"

_diagnostics, 2023, doi:10.3390/diagnostics13030503_

Round 1

Reviewer 1 Report

The present study aimed to investigate the association of FLI  with new-onset T2D in patients initiating statin therapy and following up for ≥ 3 years.

Overall the paper has a good-quality presentation. The title and abstract cover the main aspect of the work. The introduction provides background and information relevant to the study. The methods are clear, and the results match the methods described. The results are relevant to patients treated with dyslipidemia treated with statins. The findings described by the authors and conclusions correlate with the results.

However, a limitation of the study was that patients were not assessed for viral hepatitis at baseline. 

Author Response

We thank the Reviewer for his/her comments. We have now added the limitation regarding the assessment for viral hepatitis at baseline (lines 220-221).

Reviewer 2 Report

The revised manuscript, entitled “Association of Fatty Liver Index with Incident Diabetes Risk in  Patients Initiating Statin-Therapy: A 6-Year Retrospective 3 Study ” by Georgia Anastasiou et al.  is interesting and generally well-written but suffers from some flaws.

1. Please include ethics committee approval number (page 6, line 180)

2. page 2 line 56: “A complete assessment of clinical and laboratory profile was performed at the baseline visit” Was it a physician's visit?

3. T2D was diagnosed by physician?

4. Please provide the experimental design as a graph

5. Page 4 line 118 “ROC curve analysis showed that FLI had a significant predictive value for new-onset T2D (C-Statistic: 0.67; 95% CI: 0.58-119 0.77, p=0.001; Figure 1).” The AUC value 0.67 indicate on a poor predictive value of FLI.  Additionally, the sensitivity and specificity values at cut-off 60  could be calculated from the ROC analysis. 

Author Response

The revised manuscript, entitled “Association of Fatty Liver Index with Incident Diabetes Risk in  Patients Initiating Statin-Therapy: A 6-Year Retrospective 3 Study ” by Georgia Anastasiou et al.  is interesting and generally well-written but suffers from some flaws.

  1. Please include ethics committee approval number (page 6, line 180)

We thank the Reviewer for his/her comments. We have added the ethics committee approval number (line 59).

  1. page 2 line 56: “A complete assessment of clinical and laboratory profile was performed at the baseline visit” Was it a physician's visit?

The reviewer is Right. It was a physician’s visit (lines 64-65).

  1. T2D was diagnosed by physician?

The Reviewer is right. Type 2 diabetes was diagnosed by a physician, and this is now clearly stated in the manuscript (line 85).

  1. Please provide the experimental design as a graph

We have now added the study flow chart (Figure 1).

  1. Page 4 line 118 “ROC curve analysis showed that FLI had a significant predictive value for new-onset T2D (C-Statistic: 0.67; 95% CI: 0.58-119 0.77, p=0.001; Figure 1).” The AUC value 0.67 indicate on a poor predictive value of FLI.  Additionally, the sensitivity and specificity values at cut-off 60 could be calculated from the ROC analysis. 

The Reviewer is right again. We now present the sensitivity and specificity values at the cut-off 60 and underline the poor predictive value of FLI (lines 27, 146-148, and 225-227).

Reviewer 3 Report

I read with great interest the paper “Association of Fatty Liver Index with Incident Diabetes Risk in Patients Initiating Statin-Therapy: A 6-Year Retrospective Studyby Anastasiou et al.

The design is fine. The article is logically divided into sections and subsections.

Comments:

1.      Introduction: NAFLD and type 2 diabetes share common pathophysiological mechanisms, with insulin resistance as the leading one. We currently know that metabolic hepatic steatosis is a phenotype that recognizes diverse and complex causes and a wide spectrum of clinical severity, as well as considerable inter-patient variability and, therefore, the NAFLD terminology used to identify the disease it is generic and includes numerous subtypes. A consensus has recently been reached among a group of experts who proposed replacing the acronym “NAFLD” with “MALFD”, “metabolic-dysfunction-associated fatty liver disease” as a more appropriate general term  (doi: 10.37349/emed.2020.00019).

2.      “Glycated haemoglobin (HbA1c) was not measured before the diagnosis of T2D according to national guidelines”, can you please reference this sentence? This is also a limitation of the study as international guidelines include glycated haemoglobin as one of the diagnostic criteria for type 2 diabetes diagnosis, thus it is possible that some diabetic patients could be excluded.

3.      Line 150: beyond the known side effect, it is also possible that patients also present a metabolic impairment that with time may evolve into type 2 diabetes onset.

4.      What about prediabetes? Was none of the patients affected?

5.      ROC analysis is well performed, though a limit is represented by the c-statistic of 0.67 which is inferior to 0.7. Please report it.

Author Response

I read with great interest the paper “Association of Fatty Liver Index with Incident Diabetes Risk in Patients Initiating Statin-Therapy: A 6-Year Retrospective Study” by Anastasiou et al.

The design is fine. The article is logically divided into sections and subsections.

Comments:

  1. Introduction: NAFLD and type 2 diabetes share common pathophysiological mechanisms, with insulin resistance as the leading one. We currently know that metabolic hepatic steatosis is a phenotype that recognizes diverse and complex causes and a wide spectrum of clinical severity, as well as considerable inter-patient variability and, therefore, the NAFLD terminology used to identify the disease it is generic and includes numerous subtypes. A consensus has recently been reached among a group of experts who proposed replacing the acronym “NAFLD” with “MALFD”, “metabolic-dysfunction-associated fatty liver disease” as a more appropriate general term  (doi: 10.37349/emed.2020.00019).

We thank Reviewer for these insightful comments. We have now included these in the Introduction section (lines 40-46).

  1. 2.      “Glycated haemoglobin (HbA1c) was not measured before the diagnosis of T2D according to national guidelines”, can you please reference this sentence? This is also a limitation of the study as international guidelines include glycated haemoglobin as one of the diagnostic criteria for type 2 diabetes diagnosis, thus it is possible that some diabetic patients could be excluded.

We have now added this study limitation and the corresponding reference (lines 76-77, 221-223, Ref 22).

  1. Line 150: beyond the known side effect, it is also possible that patients also present a metabolic impairment that with time may evolve into type 2 diabetes onset.

The reviewer is Right. We now present more data on the independent association of NAFLD with T2D (lines 196-203, Refs 5 and 42).

  1. What about prediabetes? Was none of the patients affected?

Thank you for this comment. We have previously published relevant data (references 15-18). We now comment on the association of prediabetes and incident T2D in the Discussion section (lines 187-194).

  1. ROC analysis is well performed, though a limit is represented by the c-statistic of 0.67 which is inferior to 0.7. Please report it.

We now underline the poor predictive value of FLI (lines 27, 146-148, 225-227).

Reviewer 4 Report

This investigation is very important because NAFLD is the most common chronic liver disease,  the hepatic component of metabolic syndrome, and is associated with new-onset T2D and atherosclerotic cardiovascular disease. It is known that liver biopsy is the gold standard, and various prognostic models such as FLI have been developed for NAFLD diagnosis. This retrospective observational study that included 1241 individuals with dyslipidemia and fol- 18 lowed-up for ≥3 years  aimed to investigate the association of FLI with new-onset T2D in patients initiating statin therapy. This is the first study to investigate the association between FLI and incident T2D risk in patients initiating statin therapy.

The study was conducted in accordance with the Declaration of Helsinki and approved by the Institutional Review Board of the University Hospital of Ioannina in Greece.

The results of this study are clearly presented in Tables and Figures.

The authors concluded that FLI is significantly and independently associated with incident T2D risk in statin-treated patients. FLI could be a useful tool for predicting new T2D in patients initiating statin therapy.

Minor Revision:

The Introduction and Discussion should be extended.

Author Response

We would like to thank you for these comments. Please note that after your request, we have added 1 more Figure and expanded the Results and Discussion sections to increase total word count (2833) and total number of Figures/Tables (n=5).

Round 2

Reviewer 2 Report

Thank you for your reply to my comments.

Reviewer 3 Report

The authors appropriately answer to all the issue I raised. The paper can now be further processed for publication.